# The Segment Matters: Probable Reassortment of Tilapia Lake Virus (TiLV) Complicates Phylogenetic Analysis and Inference of Geographical Origin of New Isolate from Bangladesh

**DOI:** 10.3390/v12030258

**Published:** 2020-02-27

**Authors:** Dominique L. Chaput, David Bass, Md. Mehedi Alam, Neaz Al Hasan, Grant D. Stentiford, Ronny van Aerle, Karen Moore, John P. Bignell, Mohammad Mahfujul Haque, Charles R. Tyler

**Affiliations:** 1Biosciences, Geoffrey Pope Building, University of Exeter, Exeter, Devon EX4 4QD, UK; 2Centre for Sustainable Aquaculture Futures, University of Exeter, Exeter, Devon EX4 4QD, UK; david.bass@cefas.co.uk (D.B.); grant.stentiford@cefas.co.uk (G.D.S.); ronny.vanaerle@cefas.co.uk (R.v.A.); 3International Centre of Excellence for Aquatic Animal Health, Centre for Environment, Fisheries and Aquaculture Science (Cefas), Weymouth Laboratory, Weymouth, Dorset DT4 8UB, UK; john.bignell@cefas.co.uk; 4Department of Aquaculture, Bangladesh Agricultural University, Mymensingh-2202, Bangladesh; imran01bau@yahoo.com (M.M.A.); neaz41119@bau.edu.bd (N.A.H.); mmhaque1974@yahoo.com (M.M.H.); 5Exeter Sequencing Service, Geoffrey Pope Building, University of Exeter, Exeter, Devon EX4 4QD, UK; K.A.Moore@exeter.ac.uk

**Keywords:** tilapia lake virus, reassortment, segmented genome, phylogenetic analysis, RNA virus

## Abstract

Tilapia lake virus (TiLV), a negative sense RNA virus with a 10 segment genome, is an emerging threat to tilapia aquaculture worldwide, with outbreaks causing over 90% mortality reported on several continents since 2014. Following a severe tilapia mortality event in July 2017, we confirmed the presence of TiLV in Bangladesh and obtained the near-complete genome of this isolate, BD-2017. Phylogenetic analysis of the concatenated 10 segment coding regions placed BD-2017 in a clade with the two isolates from Thailand, separate from the Israeli and South American isolates. However, phylogenetic analysis of individual segments gave conflicting results, sometimes clustering BD-2017 with one of the Israeli isolates, and splitting pairs of isolates from the same region. By comparing patterns of topological difference among segments of quartets of isolates, we showed that TiLV likely has a history of reassortment. Segments 5 and 6, in particular, appear to have undergone a relatively recent reassortment event involving Ecuador isolate EC-2012 and Israel isolate Til-4-2011. The phylogeny of TiLV isolates therefore depends on the segment sequenced. Our findings illustrate the need to exercise caution when using phylogenetic analysis to infer geographic origin and track the movement of TiLV, and we recommend using whole genomes wherever possible.

## 1. Introduction

Tilapia are the second most important farmed finfish worldwide due to their affordability, protein and micronutrient content, resistance to disease and tolerance of high-density aquaculture conditions [1]. In Bangladesh, tilapia farming has grown markedly over the last 20 years, with Bangladesh (in 2015) ranking fourth worldwide for tilapia production [1]. Nile tilapia (*Oreochromis niloticus*) is the second most important farmed species in Bangladesh, contributing 16.6% of total aquaculture production [2] and providing a major food supply, income and employment generation for millions of poor people in rural and urban areas. Farming of this species is generally characterised by earthen pond husbandry systems, with high stocking densities, intensive feeding, and drug and chemical use for water and disease treatment [3,4,5]. In the absence of local and international regulatory frameworks of aquaculture, tilapia farmers do not follow any Better Management Practices (BMPs) and with the increasing growth of tilapia aquaculture in uncontrolled systems, disease is an emerging issue threatening the sector in Bangladesh.

As early as 2009, huge losses of cultured tilapia were reported in Israel, and the causative agent, tilapia lake virus (TiLV), was formally identified in 2014 [6]. TiLV is an enveloped virus with a 10 segment, negative-sense RNA genome [7] encoding 14 predicted proteins [8], initially proposed to belong to the family Orthomyxoviridae due to similarities in the structure of its segment termini [7], but subsequently placed in a new family, Amnoonviridae, in the same order as the Orthomyxoviridae (Articulavirales) [9]. It appears to infect tilapia at all growth stages, including fertilised eggs, egg yolk larvae, fry and fingerlings [10]. Reported mortality rates in aquaculture ponds range from as low as 5% in Egypt [11] to 90% in Thailand [12], and the virus has also been detected in asymptomatic fish with no associated mortality events [13,14,15]. Generally, however, high mortality rates are most characteristic of TiLV outbreaks [1].

Since the formal identification of TiLV in Israel [6], the virus has been reported in numerous countries across the world, including Ecuador [16], Colombia [17], Peru [14], Egypt [11,18], Tanzania and Uganda [13], India [19], Thailand [12,20], and Malaysia [21]. An assay of historical samples from tilapia hatcheries in Thailand, dating back to 2012, along with an analysis of export records, led Dong, Ataguba, et al. (2017) to issue an advisory to 40 countries, including Bangladesh, that were deemed to be at high risk of TiLV infection due to the import of potentially infected materials from these hatcheries during the study period. Determining the provenance of TiLV and tracking its movement across borders are crucial for minimising its impact on farmed and wild fish populations. However, any method used to determine how a virus arrived in a new location must be robust, as such conclusions can have significant political and economic consequences relating to trade. Phylogenetic analysis can reconstruct how pathogens have dispersed across the globe, and inferences about the movement of TiLV have been made based on phylogenetic analysis of short sequences from a single segment [18,22]. Whether this is appropriate for TiLV, with its segmented, negative-sense RNA genome that may be prone to reassortment, does not appear to have been tested, though previous studies with limited sequence data noted an absence of reassortment in this virus [14,18].

We investigated a significant tilapia mortality event in Trishal Upazila, in the Mymensingh District of Bangladesh, in July 2017. Although, at the time, TiLV had not been formally reported in Bangladesh, the characteristics of this outbreak strongly suggested a viral aetiology. We aimed to determine whether TiLV was the causative agent, and if so, to obtain the full genome and carry out phylogenetic analysis to propose how this virus arrived in Bangladesh.

## 2. Materials and Methods

### 2.1. Site Description and Sample Collection

In response to reports of high tilapia mortality in a village in Trishal Upazila, Mymensingh District, we visited one of the affected farms on 5–6 July 2017. The farmer reported that over the previous 20 days, 15 tonnes of tilapia had been lost across a 28 hectare farm. An unusually high mortality in carp had also been observed over this period, but the cultured pangasius catfish appeared to have been unaffected. At the onset of this mortality event, the farmer had treated the pond water with bleaching powder, salt and Oxy Flow (10% H_2_O_2_, Novartis Pharmaceuticals Ltd., Dhaka, Bangladesh), with no noticeable improvement in fish survivorship.

We were provided with samples from one of the affected ponds (24.5397 N 90.3445 E). This large pond (715 decimal, i.e., 2.90 hectare) was stocked predominantly with pangasius (70%), with the remaining 30% of fish consisting of Nile tilapia and carp (including rohu and common carp). Physical/chemical parameters of the pond water were all within the range commonly observed in healthy pangasius and tilapia ponds in the Mymensingh district (Appendix A).

Seven fish from the same affected pond were provided by the farm owner, terminated via Schedule 1 process, and dissected onsite: three diseased tilapia, one asymptomatic tilapia, one asymptomatic pangasius (*Pangasius bocourti*), one asymptomatic rohu (*Labeo rohita*) and one diseased common carp (*Cyprinus carpio*) (Appendix A). External symptoms of diseased tilapia included skin lesions, fin rot and sunken eyes (Appendix A), whereas diseased carp presented with swollen abdomens. Upon dissection, diseased tilapia showed small pale livers, blackish coloured gills, and small, empty stomachs. Eight tissues were sampled from each fish: heart, liver, spleen, kidney, gill, gut, gonad and skin/muscle. Samples for histology were fixed and analysed as described in the Histopathology section of the Appendix A, and samples for molecular analyses were stored in RNAlater (Ambion Inc., Austin TX, USA). All tissue samples were kept at ambient temperature until arrival in the UK, at which point those for molecular analyses were stored at −20 °C.

### 2.2. RNA Extraction and Detection of TiLV by RT-PCR

Total RNA was extracted from < 20 mg subsamples of fish tissues using the RNeasy Mini kit (Qiagen, Venlo, Netherlands), following the manufacturer’s protocol for RNAlater-fixed animal tissues (see Appendix A). RNA was eluted in 50 μL RNase-free water, quantified by spectrophotometry on a NanoDrop ND-1000 (NanoDrop Technologies Inc, Wilmington DE, USA), and stored at −80 °C.

A reverse transcription (RT)-PCR assay targeting a fragment of segment 3 was carried out to detect TiLV nucleic acids, using the semi-nested approach of Dong et al. 2017 [12], which was a modification of the nested protocol of Kembou Tsofack et al. 2017 [17] to avoid amplification of fish genes. RNA was treated with RQ1 RNase-free DNase (Promega, Madison WI, USA) and reverse transcribed, followed by two rounds of PCR and agarose gel electrophoresis. Representative amplicons were gel-purified and sequenced to confirm they were from TiLV, and short fragments of the remaining nine segments were also amplified and sequenced (full details in the Appendix A).

### 2.3. Total RNA Sequencing and TiLV Genome Assembly

A library was prepared from 100 ng total RNA from TiLV-positive F2 liver without any enrichment of mRNA or rRNA depletion, using TruSeq RNA library preparation according to the manufacturer’s protocol (Illumina Inc., San Diego CA, USA). Library was amplified and barcoded using 15 cycles of PCR. Library quality and quantity were checked using a D1000 screentape (Agilent Technologies Inc., Santa Clara CA, USA) and qPCR (New England Biolabs, Ipswich MA, USA), and 300 bp paired-end sequencing was carried out on the MiSeq with Illumina v3 SBS reagents.

Fastq files were obtained using the Illumina BCL to FASTQ file converter bcl2fastq v2.19.1.403 (Illumina Inc., San Diego CA, USA). The fastq sequences were trimmed using Trim Galore! v0.4.0 (https://www.bioinformatics.babraham.ac.uk/projects/trim_galore/) to remove sequencing adapters and low-quality bases (<Q20), and reads shorter than 150 nt were discarded. Quality-trimmed reads were normalised and error-corrected using BBNorm, part of BBTools v38.03 [23] (parameters used: ecc=t, bits=16, prefilter). In order to remove contaminating ribosomal RNA (rRNA) sequences, reads were mapped to the SILVA rRNA database (SILVA_138_SSURef_tax_silva) using Bowtie2 v2.3.5 [24] (default settings and the --very-sensitive-local parameter). Unmapped paired reads were extracted and assembled de novo using both rnaSPAdes v3.13.1 [25] (default settings and the --only-assembler parameter) and Trinity v2.8.5 [26] (default settings and the --no_normalize_reads parameter). Resulting assembled contigs were blasted against the RT-PCR amplicon sequences generated above and sequences with significant hits (E value < 1 × 10^-10^) were extracted. For each genome segment, a consensus sequence was created from the rnaSPAdes and Trinity contigs using CAP3 version date 02/10/15 [27]. The quality-trimmed paired reads were mapped to the assembled TiLV genome using bwa v0.7.17-r1188 [28] (default settings), and the results were sorted and indexed using samtools v1.9 [29], prior to statistical analysis and visualisation using QualiMap v.2.2.2 [30].

### 2.4. Phylogenetic Analyses

All available reference TiLV sequences (full and partial) were downloaded from NCBI GenBank in September 2019. At the time of writing, six complete or near-complete TiLV genomes were publicly available: two from Israel, Til-4-2011 (KU751814-823 [7]) and AD-2016 (KU552131-142); two from Thailand, TV1 (KX631921–936 [20]) and WVL18053-01A (MH319378-387 [31]); one from Ecuador, EC-2012 (MK392372-381 [32]); and one from Peru, F3-4 (MK425010-019 [14]). These were used to construct alignments of the full coding region of each of the ten segments. As some of the references were generated by PCR amplification from the start and stop codons, the segment termini were not included. We ran phylogenetic analyses on the coding region of each segment individually, and also concatenated the ten segment coding regions to obtain a multi-locus alignment.

In addition to the ten full coding region alignments with these six reference genomes, we also generated partial-segment alignments for segments 1, 2, 3, 4 and 9, for which there were additional reference sequences available. These alignments were trimmed to the consensus region of all references. Sequence alignments and trimming were carried out in AliView 1.18 [33] using Muscle [34]. Nucleotide alignments were translated to predicted amino acids using the standard table.

In MEGA7 [35], each trimmed alignment (full and partial coding regions as well as concatenated multi-locus, nucleotide and amino acid) was run through the model selection test to choose the most appropriate model of nucleotide or amino acid substitution. Overall mean distance of each full coding region alignment (nucleotide and amino acid) was computed with the selected model, using 1000 bootstrap replicates to estimate variance. Maximum likelihood phylogenetic trees were computed with the selected model, using 1000 bootstrap replicates to determine node support. Trees were viewed and annotated using the Interactive Tree of Life (iTOL) v4 [36].

### 2.5. Quartet Tree Analysis to Detect Reassortment

To look for evidence of reassortment among the TiLV segments, we used the phylogenetic approach developed by Suzuki 2010 [37] for human influenza A viruses, which is appropriate even when the main phylogenetic tree is unreliable, has unresolved nodes, or is unrooted. Using the seven TiLV isolates for which we had the full sequence of all ten segments, we selected every possible quartet (i.e., group of four isolates), 35 in total. For each quartet, we constructed ten neighbour-joining trees (one per segment) using the p-distance method with 1000 bootstrap replicates. A quartet can have only three resolved topologies (e.g., pairs A/B + C/D, or A/C + B/D, or A/D + B/C), and so we then compared the tree topologies of all ten segments for each quartet and summarised them as a string of digits, representing segments 1–10, showing where the topology differed. For example, 1111221111 indicates that all segments had the same topology except segments 5 and 6, which had a different topology (but were identical to each other). While Suzuki 2010 [37] used only quartets for which all segments trees had bootstrap support above 95%, they had whole genome sequence data for 782 H1N1 and 1663 H3N2 strains and analysed 10 million quartets, and so could afford to discard data that fell below this threshold. With seven TiLV genomes, we had only 35 quartets, and so we did not apply a strict bootstrap cut-off, and as a result, our analysis is less stringent and more limited.

### 2.6. Data Availability

The raw RNA-seq data are available in the NCBI Sequence Read Archive under BioProject number PRJNA604966. The assembled genome sequence of Bangladesh TiLV isolate BD-2017 is available in NCBI GenBank, accession numbers MN939372-MN939381 (segments 1–10). QualiMap output from genome assembly is Appendix A and on Figshare, 10.6084/m9.figshare.11812143. Multiple sequence alignments of partial segments 1, 2, 3, 4 and 9 are available as Appendix A and on Figshare, doi:10.6084/m9.figshare.11617563. Multiple sequence alignments of full coding regions of all ten segments are available as Appendix A and on Figshare, doi:10.6084/m9.figshare.11617545. Results from each quartet analysis are available as Appendix A and on Figshare, doi:10.6084/m9.figshare.11625774.

## 3. Results and Discussion

### 3.1. Detection of TiLV Nucleic Acids in Bangladesh Samples

All three diseased tilapia (F1, F2 and F7) tested positive for TiLV nucleic acids in the semi-nested RT-PCR assay (Appendix A), confirmed by Sanger sequencing of representative amplicons, which yielded identical sequences from all three fish. Histopathology findings were consistent with previous descriptions of TiLV disease (see Appendix A). F7 exhibited moderate pathognomonic signs, with a single 250 bp RT-PCR product in spleen, liver, heart and kidney samples, but no detectable TiLV nucleic acids in the other tissues. F1 and F2 showed signs of severe systemic infection, with all eight tissues yielding amplicons consistent with TiLV, and most of those exhibiting the double band pattern characteristic of heavy viral loading [12]. Notification of the presence of TiLV was made to the appropriate Competent Authority in Bangladesh.

The asymptomatic tilapia (F6) yielded no amplicons consistent with TiLV in any of the tissues most often associated with this virus (liver, spleen, kidney, heart, gill); the only detectable TiLV signal came from the gut sample (Appendix A). Whether this indicates an early stage of infection via the gastrointestinal tract or simply reflects ingestion of TiLV nucleic acids from the pond environment without true infection of gut tissue could not be determined.

The asymptomatic rohu and the diseased common carp had no detectable TiLV nucleic acids in any tissue (Appendix A), suggesting that the carp mortality observed in this pond had a different aetiology. The asymptomatic pangasius was also free of detectable TiLV nucleic acids, except for a faint RT-PCR product detected in the skin/muscle sample (Appendix A). TiLV does not generally infect other species [38], though it has been shown to cause disease in giant gourami (*Osphronemus goramy*) in Thailand [38], and has also been detected by PCR in river barb (*Puntius schwanenfeldii*) in Malaysia [39]. Further work is needed to verify whether our pangasius result indicates a true infection of pangasius skin/muscle tissue, or whether it arose from adsorption of viral particles to the outside of the fish from the pond environment or contamination from the processing of the TiLV-infected tilapia, for example via the dissection board used at the field site.

### 3.2. Bangladesh TiLV Genome and Similarity with Other TiLV Isolates

Sequencing of RNA from the liver of fish F2 on the Illumina MiSeq platform yielded enough viral sequence fragments to assemble the complete open reading frames and most of the segment termini sequences of all ten TiLV genome segments, showing that when infection is severe, it is not necessary to culture or concentrate viral particles prior to high-throughput sequencing. The ends of the segment termini were not included in the final genome sequence, as their assembly was complicated by the incorporation of host and ribosomal reads and would likely need additional sequencing by RACE PCR (rapid amplification of cDNA ends) or on a long-read platform such as PacBio. A total of 8,427,701 Illumina read pairs were generated and after quality-trimming and ribosomal RNA removal, 8,360,301 and 7,027,406 read pairs remained, respectively. Read normalisation (i.e., the removal of identical sequences, leaving only unique read pairs) reduced the number of read pairs further to 48,424 and de novo assembly of these sequences resulted in a total genome length of 10,123 nt, with an overall genome coverage of 150 ± 96, ranging from 46 ± 18 for BDS4 to 341 ± 146 for BDS9 (Table 1). A total of 8361 reads (0.06% of the reads used) mapped back to the genome (QualiMap output file available on Figshare).

Phylogenetic analyses were conducted on a large section of the genome (8871 nt; Table 1) after trimming to the consensus length of the six isolates for which the full coding regions of all ten segments were available in the NCBI GenBank database at the time of writing.

Overall mean distance analysis of the seven isolates with full coding region sequence data illustrated that some segments are more variable than others, with segments S8 to S10 being more conserved at the nucleotide level (Figure 1). Amino acid variability across the segments did not follow the same pattern as nucleotide variability; segment 9 had among the highest amino acid variability, whereas segment 1, most diverse at the nucleotide level, had among the lowest amino acid variability, pointing to a higher proportion of silent mutations in that segment. In general, amino acid variability was too low to be useful for phylogenetic analysis, yielding poor resolution and node support for most segments (Appendix A).

The maximum likelihood phylogenetic tree based on the concatenated multi-locus alignment of all ten segment coding regions grouped the Bangladesh TiLV isolate BD-2017 with the two Thailand references (TV1 and WVL18053-01A), whereas the two South American strains (Peru F3-4 and Ecuador EC-2012) clustered together with Israel Til-4-2011 (Israel AD-2016 could not be resolved) (Figure 2). This branching pattern resembles the Thai and Israeli clades identified by Pulido et al. 2019 [14], though their analysis was able to include AD-2016 in the Israeli clade. Given the likely import into Bangladesh of TiLV-infected materials from Thai hatcheries between 2012 and 2017 [10], the similarity of the Bangladesh strain to the two Thai references is not surprising.

Next, we attempted to assess the similarity of BD-2017 to isolates from additional countries, as there were many more partial sequences in GenBank than the six isolates used for the concatenated tree in Figure 2. To date, there has not been any consensus on the TiLV segment used for identification and phylogenetic analyses, which makes it difficult to assess how isolates from different countries are related. This would need to be clarified if TiLV is eventually to be listed as a notifiable disease by the OIE. A short portion of segment 3 has been most widely sequenced, as it is the target of the semi-nested RT-PCR assay used for TiLV detection [12,17]. Sections of segments 1, 2, 4 and 9 have also been used. We therefore constructed separate phylogenetic trees for segments 1, 2, 3, 4 and 9 (Figure 3). Most of these alignments (except for segment 1) were shorter than the full coding region, as they were trimmed to the consensus of all references available in GenBank at the time of writing.

Compared with the topology of the concatenated tree in Figure 2, when these shorter sequence fragments from additional isolates were included in phylogenetic analyses, branching patterns generally changed, and both resolution and node support tended to decrease (Figure 3). The exception was segment 1 (Figure 3a), whose topology mirrored the concatenated tree, though unlike the other trees, segment 1 was based on the whole segment coding region (1560 nt) rather than a short fragment, and all the additional references were from Thailand and formed a distinct clade that also included the Bangladesh TiLV. The other trees with additional references (segments 2, 3, 4, 9) showed a different branching pattern to the concatenated tree. With segment 2, two African strains (from Tanzania and Uganda) clustered with Israel AD-2016, while the other Israel strain formed a cluster with the two South American and the Bangladesh strain. Segment 3 formed some regional clusters (South America, India, Egypt), but it split the Thai sequences, and formed three separate Israel clusters (though did not distinguish the two clades identified by Skornik et al. 2019 [22]). Furthermore, the relationships between the regions could not generally be resolved. Segment 4 showed poor resolution and node support. The sequences from Egypt and India formed two distinct clusters, but the other strains could not be resolved. Segment 9 also showed poor resolution and node support, splitting the South American strains.

### 3.3. Possible Reassortment of TiLV Segments

Since the clustering pattern observed with the concatenated tree of the coding regions from all ten segments (Figure 2) did not hold when shorter fragments of segments 2, 3, 4 and 9 were analysed with additional isolates (Figure 3), we constructed individual maximum likelihood trees for all ten segments using only the seven references in the concatenated tree (Figure 4), for which the full coding region on each of the ten segments was available. We expected all the segment trees to reflect the topology of the concatenated tree, since they were constructed from the entire coding regions and not from much shorter fragments.

However, while some segments, notably 2 and 7, showed clustering that reflected the whole genome pattern (albeit segment 7 had poorer resolution and node support), other segments exhibited different branching patterns with high node support (Figure 4). With segment 1, the Bangladesh isolate BD-2017 clustered not with Thai references but rather with Israel AD-2016. With segment 4, Thailand TV1 clustered with both Israel isolates, but with segment 5, the two Thailand isolates were separated, while BD-2017 clustered with the Israel and South American isolates. With both segment 5 and 6, Israel Til-4-2011 was grouped with Ecuador EC-2012, and with segment 9, the South American and Thai clades were both split, with Bangladesh TiLV grouped with Thailand WVL18053-01A.

These contrasting phylogenetic patterns for different segments suggest that TiLV has undergone reassortment, a phenomenon that can occur in viruses with segmented genomes, whereby co-infection of a host cell by two (or more) strains results in progeny viruses with genome segments from different parent strains [40,41]. While most reassortment events are thought to reduce the fitness of progeny viruses, occasionally they can result in novel strains with increased virulence [41]; indeed, reassortment of influenza A has resulted in new strains that have caused global flu pandemics, notably the 1957 ‘Asian’, 1968 ‘Hong Kong’, and 2009 ‘swine flu’ pandemics [40,42]. Influenza viruses are in the family Orthomyxoviridae, to which TiLV shows some structural similarities [7]; however, TiLV belongs to the Amnoonviridae [9], and earlier studies have noted an absence of reassortment based on the limited sequence data available at the time [14,18].

The individual segment trees (Figure 4) suggested reassortment, but the small number of genomes available (seven) and the poor resolution of some of the segment trees made it difficult to test this hypothesis. Methods to test for reassortment of segmented genomes generally require well-resolved, rooted phylogenetic trees with far more reference sequences than were available for TiLV. We selected the method devised by Suzuki 2010 [37], based on the analysis of quartets (i.e., groups of four isolates) and the comparison of tree topologies among the different segments. This approach is suitable when trees are poorly resolved and unrooted, since quartets can have only one of three resolved tree topologies, which facilitates comparison among segments.

From seven isolates, 35 unique quartets can be selected. For each of these 35 quartets, we computed neighbour-joining trees using p-distance and 1000 bootstrap replicates for all ten segments, though only segments 1–7 were included in subsequent analyses, as segments 8–10 are too conserved and resulted in low bootstrap support for most quartets (Appendix A). With 17 of the quartets, each segment had the same tree topology (Table 2), represented as the string of characters 1111111. This indicates that the phylogenetic relationship of the four isolates in each of these quartets is the same for all seven segments. There is no evidence of reassortment between these isolates.

However, the next two most abundant patterns, 1111221 and 2211221 (5 and 4 quartets, respectively), point to a reassortment of segments 5 and 6, and the latter pattern also suggests a reassortment of segments 1 and 2 (Table 2). All these quartets included Israel Til-4-2011 and one or both of the South American isolates (Ecuador/Peru). The segments with a contrasting tree topology (indicated by the character ‘2’) consistently grouped Til-4-2011 with Ecuador EC-2012 (or Peru F3-4 when the Ecuador isolate was not present in the quartet), whereas the other segments of those same quartets (indicated by the character ‘1’) clearly separated Til-4-2011 from the South American isolates. Bootstrap values were generally high (>90%) for most segments in these quartets (Table 3). Furthermore, pairwise sequence comparisons showed that segments 5 and 6 of Ecuador EC-2012 are 100% identical to those of Israel Til-4-2011, but no other pairs of isolates have identical sequences for any of the segments. Despite the fact they were identified on different continents, the history of these two isolates therefore appears to include a relatively recent reassortment of segments 5 and 6. The reassortment of segments 1 and 2 is less clear, and more whole-genome sequences from additional isolates are required to confirm this.

### 3.4. Amino Acid Substitutions

The reported severity of TiLV varies widely; in some locations, it has been detected in asymptomatic fish only through screening with molecular methods, with no associated mortality (Peru, Tanzania, Uganda), whereas in other locations, it has caused rapid, severe outbreaks with over 90% mortality. Nucleotide-based phylogenetic analysis does not appear to explain this variability in mortality rates, as extremely lethal strains (e.g., Ecuador EC-2012) cluster with asymptomatic ones (e.g., Peru F3-4) based on concatenated alignments (Figure 2). It is possible that reassortment explains some of this variability in virulence. For example, we have shown that Ecuador EC-2012 has reassorted with the lethal type strain Israel Til-4-2011, but Peru F3-4 has not, and reassortment in other segmented RNA viruses, notably influenza, is known to have generated highly virulent progeny strains [40,41].

However, most nucleotide substitutions in TiLV alignments are silent mutations (i.e., they do not result in an amino acid change, and therefore do not affect protein structure and function), and some TiLV strains that clustered into separate lineages at the nucleotide level have segments with identical amino acid sequences (Figure 5a,b). The Bangladesh BD-2017 segment 2 amino acid sequence is identical to that of Ecuador EC-2012. Segment 3 of Ecuador EC-2012 is identical to the Israel type strain, as are both the segment 4 amino acid sequences of two of the Egypt strains (Farm3 and Farm5).

For functional changes, amino acid sequence alignments are more informative than nucleotide alignments, but since TiLV amino acid sequence alignments are not sufficiently variable to allow robust phylogenetic analysis, we instead examined patterns of amino acid substitutions in each of the ten segments to look for specific mutations that could explain the variation in TiLV disease severity (Figure 5a,b), since some amino acid substitutions have greater impact on protein structure and function than others (e.g., when a hydrophilic amino acid is replaced by a hydrophobic one).

We found no obvious pattern linking amino acid substitutions and outbreak severity (Figure 5a,b), though this analysis was hampered by a lack of disease severity metadata for many of the reference sequences. Furthermore, only one of the isolates with full coding region sequences and severity data, Peru F3-4, was not from a severe outbreak, and so we had few examples of asymptomatic or low-severity strains. The Peru F3-4 isolate, which was asymptomatic, did have differences in several amino acids from the type strain in most segments, but other isolates from severe outbreaks had similar numbers of substitutions, with similarly high predicted impact on protein structure and function, notably EC-2012 and TV1. The only suggested link between specific amino acid changes and severity came from the set of segment 3 sequences from Israel [22], which all had severity information. A change from lysine (K) to arginine (R) at position 127 appears only in moderate strains and not in any severe strain. However, this change is between two positively charged hydrophilic amino acids, and so is not predicted to have a large impact on protein structure and function, and it is not present in all moderate strains from that study.

Some TiLV segments appear to tolerate considerable amino acid changes while maintaining virulence. For example, in segment 6, the Bangladesh and Thailand TV1 isolates have several major substitutions compared with the Israel type strain, including a change from the hydrophilic, positively charged lysine (K) to the hydrophobic isoleucine (I) at position 55, and from the hydrophobic methionine (M) to the positively charged hydrophilic arginine (R) (position 95) or lysine (K) (position 266). Other segments, notably segment 1, are largely devoid of major mutations despite being much longer, which suggests that the translated protein is especially sensitive to amino acid changes. Segment 1 encodes a protein with weak homology to the influenza C polymerase PB1 subunit [7], and so is thought to encode the TiLV RNA polymerase, responsible for viral replication and transcription. Although in silico analysis predicted fourteen TiLV proteins in total, six of which have transmembrane helix regions [8], the proteins encoded by segments 2–10 have no relatives in reference databases, and so their functions are unknown [7].

## 4. Conclusions

We confirmed the presence of TiLV causing disease in tilapia in Bangladesh and obtained the near-complete genome sequence of this isolate, BD-2017, including the full coding regions of all ten segments and most of the segment termini. While phylogenetic analysis based on a short fragment of one or a few segments is often used to infer how TiLV has moved between geographical areas [18,22], we have shown that TiLV segments have different degrees of nucleotide and amino acid variation, and that TiLV has a history of genome reassortment, notably involving segments 5 and 6 (but perhaps also segments 1 and 2). Consequently, such conclusions depend largely on which TiLV segment is sequenced. We therefore advise caution when using phylogenetic analysis to trace the origin and track the movement of this virus between countries, and we recommend using whole-genome sequencing whenever possible.

TiLV is a not a human health risk but it has huge potential impacts on global food security and nutrition. It varies widely in severity (from asymptomatic to extremely lethal) for reasons that are currently unresolved, but reassortment may be a contributing factor, since, in other segmented RNA viruses such as influenza, reassortment has resulted in the sudden emergence of extremely virulent strains.

FAO issued an alert in 2017 for tilapia-producing countries to be vigilant and initiate an active TiLV surveillance programme [1]. Tilapia-producing countries are also encouraged to launch public information campaigns to advise tilapia producers, who tend to be small-scale farmers, of the threat posed by TiLV, its clinical signs, potential impacts, and the need to report mortality events to authorities [1]. The information derived from our study will be useful for government and research organizations to initiate a TiLV surveillance programme in Bangladesh, and for launching information campaigns for the farmers and other value-chain stakeholders.

## Figures and Tables

**Figure 1 viruses-12-00258-f001:**
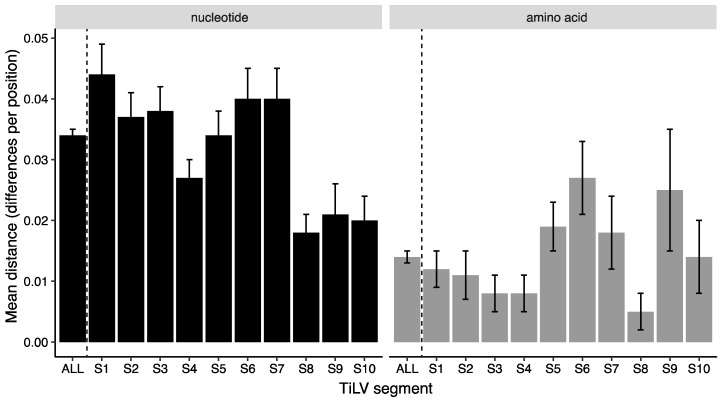
Mean TiLV nucleotide and amino acid distance (differences per position), based on alignment of the full coding regions of seven genomes from Israel, Thailand, Ecuador, Peru and Bangladesh. Error bars show estimated variance based on 1000 bootstrap replicates, and models of nucleotide substitution for each segment are as listed in Figure 4.

**Figure 2 viruses-12-00258-f002:**
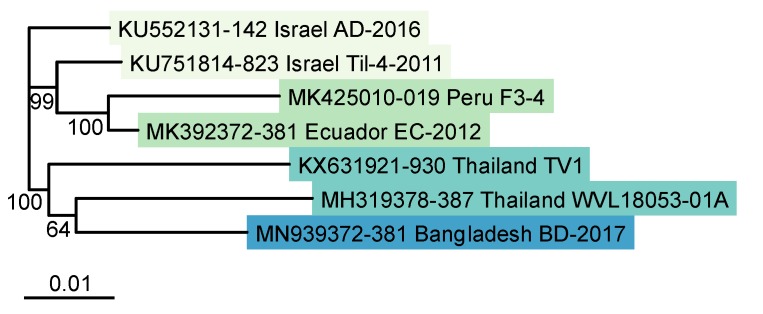
Maximum likelihood phylogenetic tree of TiLV genomes (concatenated coding regions from segments 1–10), based on 8871 nucleotide positions, using the Tamura Nei model of nucleotide substitution with gamma-distributed rate heterogeneity (TN93+G). Bootstrap values were calculated from 1000 replicates, and percent support is shown on nodes where values exceed 50%. Scale bar shows number of substitutions per site.

**Figure 3 viruses-12-00258-f003:**
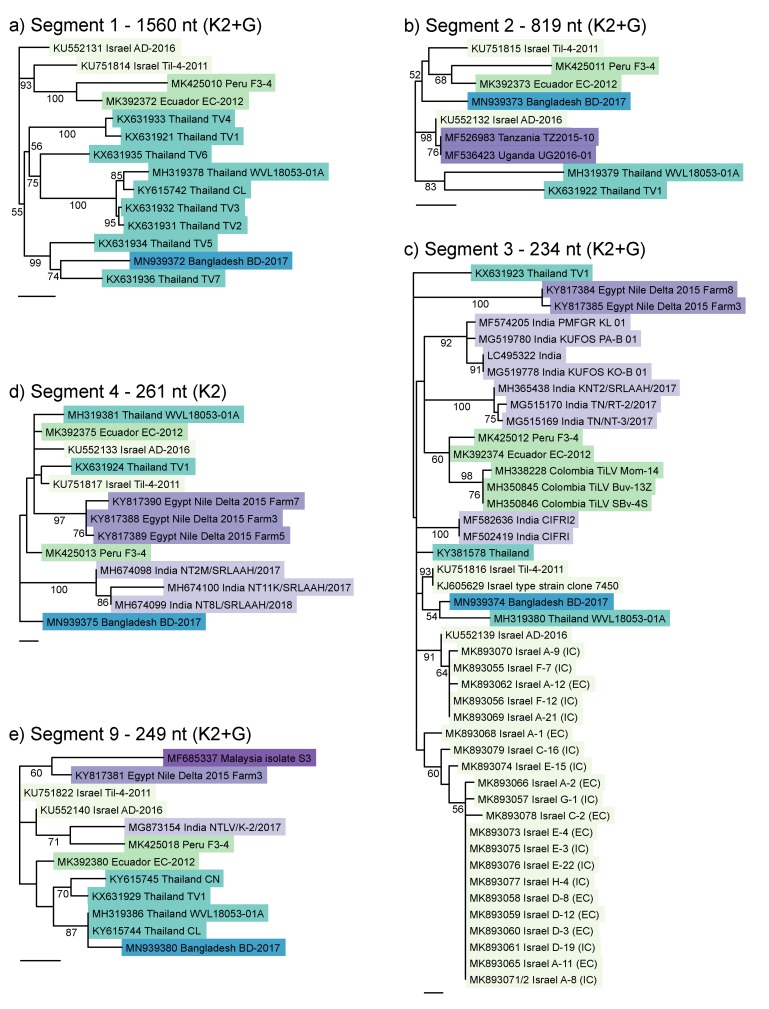
Maximum likelihood phylogenetic trees of the five TiLV segments for which additional reference sequences were available in GenBank. (**a**) Segment 1; (**b**) Segment 2; (**c**) Segment 3; (**d**) Segment 4; (**e**) Segment 9. Alignment lengths and best-fitting nucleotide substitution models are shown in panel headings (K2 = Kimura 2 parameter model, +G = gamma-distributed rate heterogeneity). Bootstrap values were calculated from 1000 replicates, and percent support is shown on nodes where values exceed 50%. Scale bars show 0.01 substitutions per site. Colours show isolates from the same region.

**Figure 4 viruses-12-00258-f004:**
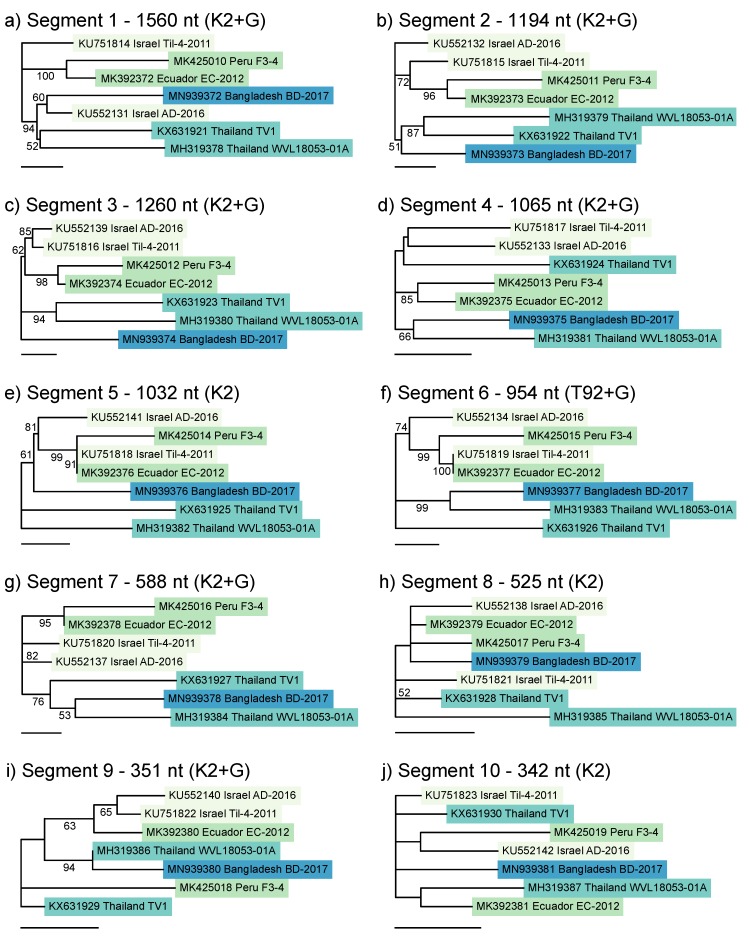
Maximum likelihood phylogenetic trees of the ten TiLV segments, using only isolates for which the full coding regions of all ten segments have been sequenced. (**a**) Segment 1; (**b**) Segment 2; (**c**) Segment 3; (**d**) Segment 4; (**e**) Segment 5; (**f**) Segment 6; (**g**) Segment 7; (**h**) Segment 8; (**i**) Segment 9; (**j**) Segment 10. Alignment lengths and best-fitting nucleotide substitution models are shown in panel headings (K2 = Kimura 2 parameter model, T92 = Tamura 1992 model, +G = gamma-distributed rate heterogeneity). Bootstrap values were calculated from 1000 replicates, and percent support is shown on nodes where values exceed 50%. Scale bars show 0.01 substitutions per site. Colours show isolates from the same region.

**Figure 5 viruses-12-00258-f005:**
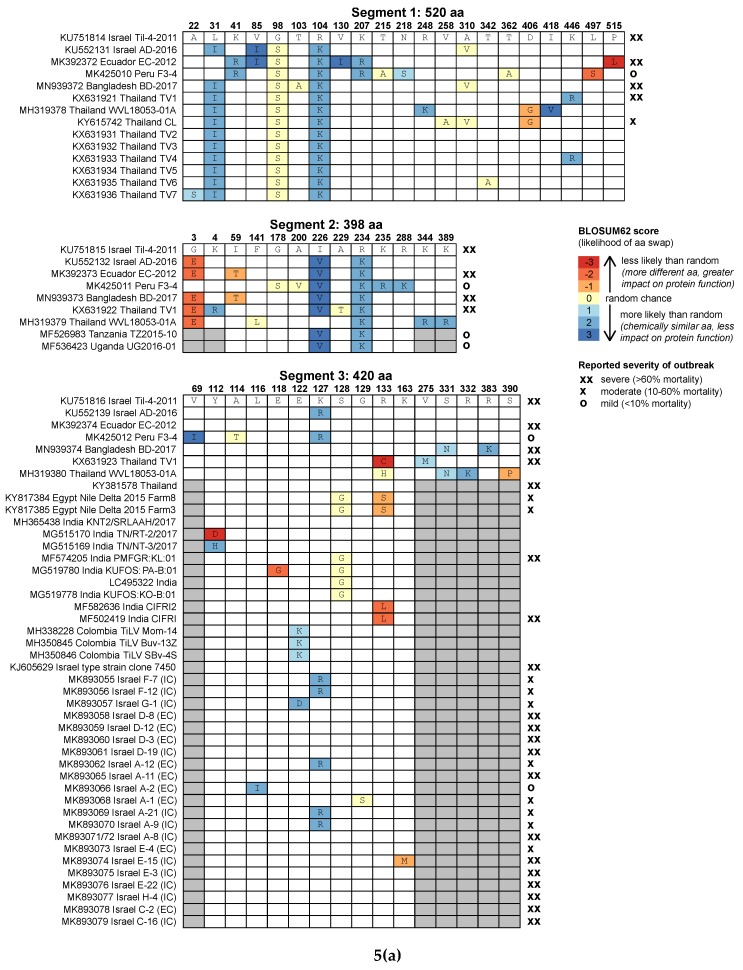
(**a**,**b**) Predicted amino acid differences in the ten TiLV segments, compared to the reference strain Til-4-2011 from Israel. Numbers along the top show the amino acid position at which there is a difference in at least one sequence, the first line shows the sequence of strain Til-4-2011, and letters in other boxes show where amino acids differ. Colours indicate the BLOSUM62 score, a measure of how frequently the amino acid difference occurs in protein alignments (with blue colours indicating more similar amino acids and red indicating more dissimilar amino acids, with a greater impact on protein structure and function). Grey boxes indicate that the sequence did not extend to that region. Symbols along the right show the severity of outbreak, where reported (xx > 60% mortality, x = 10%–60% mortality, o < 10% mortality or asymptomatic, and no symbol = data not available).

**Table 1 viruses-12-00258-t001:** Length and coverage of the Tilapia lake virus (TiLV) segments from the Bangladesh isolate, and length of sequences used for phylogenetic analyses.

Segment	Reported Length (nt) from Bacharach et al. [7]	BD-2017 Length (nt)	BD-2017 Coverage (Mean ± SD)	Trimmed Length with Full Coding Region Refs (nt)	Partial Segments with Shorter Refs (nt)
1	1641	1620	90 ± 29	1560	1560
2	1471	1448	91 ± 24	1194	819
3	1371	1353	204 ± 44	1260	234
4	1250	1226	46 ± 18	1065	261
5	1099	1083	198 ± 61	1032	
6	1044	1024	177 ± 50	954	
7	777	758	92 ± 31	588	
8	657	637	289 ± 72	525	
9	548	531	341 ± 146	351	249
10	465	443	164 ± 44	342	
Total	10323	10123	150 ± 96	8871	

**Table 2 viruses-12-00258-t002:** Topological patterns of the first seven TiLV genomic segments, and distribution of the 35 possible quartets among the observed patterns.

Pattern (Segs 1–7)	Number of Quartets	Notes
1111111	17	Identical pattern for all segments—no reassortment
1111221	5	1 = Ecuador/Peru together and/or both Israel together,
2211221	4	2 = Ecuador with Israel Til-4-2011
xx11111	2	Segs 1–2 poorly resolved, 3–7 well supported
unresolved	7	Most/all segments poorly resolved

**Table 3 viruses-12-00258-t003:** Examples of quartets showing topological patterns that suggest reassortment of TiLV segments. Topologies are from neighbour-joining trees of each segment, with more closely related pairs listed. Bootstrap values were obtained from 1000 resamplings.

Segment	Bootstrap Support (%)	Pair 1	Pair 2
Pattern 1111221			
1	100	Ecuador + Peru	Israel AD-2016 + Israel Til-4-2011
2	98	Ecuador + Peru	Israel AD-2016 + Israel Til-4-2011
3	99	Ecuador + Peru	Israel AD-2016 + Israel Til-4-2011
4	95	Ecuador + Peru	Israel AD-2016 + Israel Til-4-2011
5	74	Ecuador + Israel Til-4-2011	Israel AD-2016 + Peru
6	97	Ecuador + Israel Til-4-2011	Israel AD-2016 + Peru
7	98	Ecuador + Peru	Israel AD-2016 + Israel Til-4-2011
Pattern 2211221			
1	95	Ecuador + Israel Til-4-2011	Israel AD-2016 + Thailand WVL
2	93	Ecuador + Israel Til-4-2011	Israel AD-2016 + Thailand WVL
3	92	Ecuador + Thailand WVL	Israel AD-2016 + Israel Til-4-2011
4	70	Ecuador + Thailand WVL	Israel AD-2016 + Israel Til-4-2011
5	100	Ecuador + Israel Til-4-2011	Israel AD-2016 + Thailand WVL
6	100	Ecuador + Israel Til-4-2011	Israel AD-2016 + Thailand WVL
7	42	Ecuador + Thailand WVL	Israel AD-2016 + Israel Til-4-2011

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
