# Peer review of "The Segment Matters: Probable Reassortment of Tilapia Lake Virus (TiLV) Complicates Phylogenetic Analysis and Inference of Geographical Origin of New Isolate from Bangladesh"

_viruses, 2020, doi:10.3390/v12030258_

Round 1

Reviewer 1 Report

This paper presents the first molecular identification of a virus from Tilapia, TiLV, in Bangladesh, and the comparison of its complete genome with other isolates of various origins. The authors found out that some genome components exhibited relations with sequences from Asia while others were more similar to viruses from South America, suggesting reassortements. Although this hypothesis would need confirmation by other studies, this is a new potential mode of evolution for devastating virus. The subject is therefore very important.

The work is well conducted and excellently presented. Some data are surprising (for instance, positive signal for pangasius), but they are appropriately discussed. I recommend to publish it after minor corrections.

  • It would have been a good idea to sequence the segment 5 and 6 of F1 and F2 to see if their groupings in the trees are the same than for F7. In other words, is there one or different strains in this outbreak ? and were the putative reassortements created before or during the outbreak. Did the authors try to obtain that data ?
  • Some recent data about the function of the (14 ?) genes should be mentioned, either in the introduction or in the results near line 438 (or both). See ‘Structural Characterization of Open Reading Frame-Encoded Functional Genes from Tilapia Lake Virus (TiLV)’ (Mol. Biotech. 2019).
  • Ref 22. Refer to the website (as recommended by the author)
  • Figure S6. Mention the organ, or make a fusion with S5.

Author Response

viruses-729319: The Segment Matters: Probable Reassortment of Tilapia Lake Virus (TiLV) Complicates Phylogenetic Analysis and Inference of Geographical Origin of New Isolate From Bangladesh

Response to Reviewer 1: We thank Reviewer 1 for their thoughtful comments and suggestions, which have allowed us to improve this manuscript. Below are our responses to the specific points raised. We have incorporated the resulting changes into the manuscript.

For clarity, in this document, reviewer comments are in italics, with original line numbers. Corresponding line numbers in the revised manuscript are specified as [revised line xx]. Our response to each comment is indented, and modified or new text that is now in the manuscript is underlined.

It would have been a good idea to sequence the segment 5 and 6 of F1 and F2 to see if their groupings in the trees are the same than for F7. In other words, is there one or different strains in this outbreak?

Response: This is a good question. Initially, we sequenced only the short fragment of segment 3 (by Sanger sequencing) generated during the RT-PCR assay, and the sequence was identical in all three affected fish (F1, F2, F7). We have added the following to include this information [revised lines 221-222]: '[...] confirmed by Sanger sequencing of representative amplicons, which yielded identical sequences from all three fish.' Since the sequence was identical in the three fish, and they had all come from the same pond, we selected the RNA from only one fish for further sequencing by RNA-seq rather than attempt to get the TiLV genome from multiple fish. We were not sure how much sequence data were required for assembly from this sample type, so did not want to multiplex multiple potentially different samples on our sequencing run. Future work should look at fine scale TiLV genome variations within single outbreaks, with whole genomes (including segments 5 and 6).

and were the putative reassortements created before or during the outbreak. Did the authors try to obtain that data?

Response: The clearest reassortment that we were able to identify was the segment 5/6 reassortment between isolates from Ecuador and Israel. There were no obvious reassortments with BD-2017 that could be identified with the relatively small number of reference sequences available. Detecting reassortment during an outbreak would be very interesting, but would require far more sequence data than we could generate within the scope of our project. Now that we and others have shown that the TiLV genome can be assembled from sequencing of tissue RNA (rather than requiring concentration of viral particles or propagation in cell cultures), it should be easier for other groups to sequence multiple samples from the same outbreak to look for reassortment events.

Some recent data about the function of the (14 ?) genes should be mentioned, either in the introduction or in the results near line 438 (or both). See ‘Structural Characterization of Open Reading Frame-Encoded Functional Genes from Tilapia Lake Virus (TiLV)’ (Mol. Biotech. 2019).

Response: We have modified the introduction as follows to include the number of proteins predicted by the Acharya et al. paper: 'TiLV is an enveloped virus with a 10-segment, negative sense RNA genome encoding 14 predicted proteins, ...' [revised lines 55-56]. We have also added the following text to the last paragraph of the Results and Discussion section, and we have referred to the Acharya et al. 2019 paper, as suggested: 'Although in silico analysis predicted fourteen TiLV proteins in total, six of which have transmembrane helix regions, the proteins encoded by segments 2-10 have no relatives in reference databases so their functions are unknown' [revised lines 489-491].

Ref 22. Refer to the website (as recommended by the author)

Response: We have added the website for this software: sourceforge.net/projects/bbmap/ [now reference 23, revised line 626].

Figure S6. Mention the organ, or make a fusion with S5.

Response: We have modified the caption to mention the organ, as suggested: 'Large multinucleated syncytia (arrows) in liver tissue often contained...'

Reviewer 2 Report

Comments to viruses-729319

A review of the manuscript entitled The Segment Matters: Probable Reassortment of Tilapia Lake Virus (TiLV) Complicates Phylogenetic Analysis and Inference of Geographical Origin of New Isolate From Bangladesh by Dominique L. Chaput, David Bass, Md. Mehedi Alam, Neaz Al Hasan, Grant D. Stentiford, Ronny van Aerle, Karen Moore, John P. Bignell, Mohammad Mahfujul Haque, and Charles R. Tyler.

The main findings of the manuscript are gene segment reassortments between strains of Tilapia Lake Virus (TiLV) separated by large geographical distances, and the need to sequence all gene segments of this virus to ensure that accurate conclusions are drawn with respect to phylogeny/epidemiology. The manuscript is comprehensive, well-structured and well written. The phylogenetic trees for individual segments, together with the quartet tree analysis and the tables displaying amino acid differences between the strains, provide good support for single and possibly multiple gene segment reassortment events. Reassortment between an Israeli vs. Ecuadorian strain seems to have particularly strong support. To my knowledge this is the first report of reassortment for this virus. It will be interesting to follow the evolutionary dynamics of this virus as more genomes are sequenced. In my opinion the paper can be published as is but could benefit from a few minor revisions outlined below.

A few issues that should be addressed:

Lines 152-155, 172 and elsewhere: Although described, the term whole-genome means the entire genome and should only be used for complete genome sequences. Goes for both the reference genomes and the concatenated genome.

Line 190: It would have been preferable to have the files that are in Figshare put into supplementary material (if possible), as Figshare seems to have an inconvenient account/user login system.

Line 222: complete open reading frame

Line 228: the term normally used in this context I believe is ribosomal RNA.

Line 229: The term read normalization does not provide much information. Although the approach or pipeline is well described in M&M I suggest rephrasing it and provide a bit more detail. An extra line or sentence should suffice.

Fig S8 legends: amino acid sequence alignments. Other places as well.

Line 289: ...compared to the whole-genome tree. Also, not whole genome as end regions are missing. Use another term (i.e. concatenated genome sequence tree, tree generated from concatenation of coding regions, or similar)

Line 427 (and others): ….have large impact on protein structure and function.

Line 557: If possible, provide more details in ref. 22.

Author Response

viruses-729319: The Segment Matters: Probable Reassortment of Tilapia Lake Virus (TiLV) Complicates Phylogenetic Analysis and Inference of Geographical Origin of New Isolate From Bangladesh

Response to Reviewer 2: We thank Reviewer 2 for their thoughtful comments and suggestions, which have allowed us to improve this manuscript. Below are our responses to the specific points raised. We have incorporated the resulting changes into the manuscript.

For clarity, in this document, reviewer comments are in italics, with original line numbers. Corresponding line numbers in the revised manuscript are specified as [revised line xx]. Our response to each comment is indented, and modified or new text that is now in the manuscript is underlined.

Lines 152-155, 172 and elsewhere: Although described, the term whole-genome means the entire genome and should only be used for complete genome sequences. Goes for both the reference genomes and the concatenated genome.

Response: This is a good point. We have changed the wording throughout the manuscript to clarify that we did not work with full genomes, but rather with full coding regions (excluding the non-coding segment termini). For example, we have changed 'whole segment' to 'full coding region', 'whole-genome reference sequences' to 'isolates for which the full coding regions of all ten segments were available', and 'whole genome tree' to 'concatenated multi-locus tree'. In the Abstract, we specify that we obtained the 'near-complete genome' rather than the genome.

Line 190: It would have been preferable to have the files that are in Figshare put into supplementary material (if possible), as Figshare seems to have an inconvenient account/user login system.

Response: We have put all the files currently on Figshare into supplementary materials. This includes the alignments used to make the phylogenetic trees, the QualiMap file from the genome assembly, and the output of the quartet analysis. We have also kept these files on Figshare, for readers who might prefer that platform.

Line 222: complete open reading frame

Response: We have changed 'full open reading frame' to 'complete open reading frame', as suggested [revised line 247].

Line 228: the term normally used in this context I believe is ribosomal RNA.

Response: We have changed 'ribosomal DNA' to 'ribosomal RNA' [revised line 254].

Line 229: The term read normalization does not provide much information. Although the approach or pipeline is well described in M&M I suggest rephrasing it and provide a bit more detail. An extra line or sentence should suffice.

Response: To clarify what we did at this step, we have added an explanation of read normalisation in parentheses: 'Read normalisation (i.e. the removal of identical sequences, leaving only unique read pairs) reduced the number...' [revised lines 254-255].

Fig S8 legends: amino acid sequence alignments. Other places as well.

Response: We have modified Fig S8 legend as suggested: '[...] based on predicted amino acid sequence alignments...' We have also modified the other places where the phrase 'amino acid alignments' occurred, so that they now read 'amino acid sequence alignments' [revised lines 460, 461, 531].

Line 289: ...compared to the whole-genome tree. Also, not whole genome as end regions are missing. Use another term (i.e. concatenated genome sequence tree, tree generated from concatenation of coding regions, or similar)

Response: As mentioned above, we have changed the wording throughout the manuscript to clarify that we are not using the whole genome. With regards to the tree, we first refer to it as the 'tree based on the concatenated multi-locus alignment of all ten segment coding regions' [revised lines 291-292], and subsequently as the 'concatenated tree' [e.g. revised lines 307, 317, 320, 324 and others].

Line 427 (and others): ….have large impact on protein structure and function.

Response: We have changed the text here from 'protein function' to 'protein structure and function', and also in the other places that phrase occurred [revised lines 438, 457, 464-465, 474, 478-479].

Line 557: If possible, provide more details in ref. 22.

Response: We have added the website for this software: sourceforge.net/projects/bbmap/ [now reference 23, revised line 626].